**Data Availability Statement:** No datasets were generated or analysed during the current study. All

# A protocol for a controlled, pre-post intervention study to change attitudes toward child marriage in Southwestern Nigeria using targeted radio programming

Olubukola Omobowale[1], Olayinka Omigbodun[2,3], Olayinka Egbokhare[4], Alissa Koski[5,6]*

1 Rehabilitative and Social Medicine Unit, Department of Community Medicine, College of Medicine, University of Ibadan, Ibadan, Nigeria, 2 Centre for Child and Adolescent Mental Health, College of Medicine, University of Ibadan, Ibadan, Nigeria, 3 Department of Psychiatry, College of Medicine, University of Ibadan, Ibadan, Nigeria, 4 Biomedical Communication Centre, College of Medicine, University of Ibadan, Ibadan, Nigeria, 5 Department of Epidemiology, Biostatistics, and Occupational Health, McGill University, Montreal, Canada, 6 Department of Equity, Ethics, and Policy, McGill University, Montreal, Canada

* alissa.koski@mcgill.ca

## Abstract

### Background

Child marriage hinders progress toward population health and development goals. Cost effective interventions that address the root causes of child marriage are needed to speed progress toward ending the practice. Nigeria is home to the largest number of married girls in Africa and many of these girls are members of the Hausa ethnic group, making efforts to tackle this issue particularly urgent among this population.

### Methods

Radio programs have the potential to inform large numbers of people about the harms of child marriage and change their support for the practice at low cost. We will develop a series of radio programs that address gender inequitable attitudes that motivate child marriage among Hausa communities in Ibadan, Nigeria. The content of the series will be based on input from the Hausa community. A baseline survey that measures knowledge of and support for child marriage will be conducted among randomly selected samples of Hausa adults in two cities: Ibadan, which will serve as the intervention site, and Akure, the control site. The radio programs will then air on Hausa-language stations in Ibadan over a three-month period, with the aim of informing persons of the potential harms of child marriage and reducing their support for the practice. A follow-up survey with the same individuals surveyed at baseline will be conducted in both cities. We will measure the impact of this intervention by comparing changes in these outcomes over time in the intervention site (Ibadan) with changes in the same outcomes in the control site (Akure).

relevant data from this study will be made available upon study completion.

**Funding:** Funding for this study comes from the Grand Challenges Canada Stars in Global Health Program and was awarded to O. Omobowale and A. Koski (Grant Number: ST-POC-2206-53579). Grand Challenges Canada https://www.grandchallenges.ca) The funders had and will not have a role in study design, data collection and analysis, decision to publish, or preparation of the manuscript.

**Competing interests:** The authors have declared that no competing interests exist

## Conclusion

This study will investigate whether a series of targeted radio programs can reduce support for child marriage. The intervention is readily scalable and cost-effective and, if it effectively shifts attitudes toward child marriage, could represent a promising way of addressing child marriage in Nigeria.

## Introduction and rationale

Child marriage, defined as marriage before 18 years of age, is a violation of human rights with harmful consequences for population health and development [1, 2]. Girls are far more likely to marry as children than boys, and so the consequences of child marriage exacerbate existing gender inequities across much of the globe. Child marriage cuts short girls' education [3, 4], increases their likelihood of living in poverty in adulthood [5], and is strongly and consistently correlated with early childbearing, lack of access to maternal health care, and intimate partner violence [2]. The prominence of child marriage on global health and development agendas is reflected by the fact that the United Nations Sustainable Development Goals call for elimination of the practice by the year 2030 [1].

Child marriage is ongoing throughout most of the world despite intensifying efforts to eliminate it [6]. At its core, the practice is driven by the belief that girls are less valuable than boys. Often, these beliefs intersect with poverty to result in child marriage. To illustrate, if a family of limited means can only afford to educate some of their children, they may see more value in sending their male children to school. In the absence of educational or economic opportunities, female children may then be married at young ages. Child marriage frequently reflects efforts to control girls' sexuality as well. Girls may be married at young ages to prevent them from engaging in consensual premarital sex or in an effort to prevent sexual violence where unmarried women are more likely to be targeted [7, 8].

Many interventions intended to curb child marriage have been fielded in recent decades. Since 2011, three systematic reviews have attempted to identify interventions that are most effective at meeting that aim [9–11]. The most recent review by Malhotra and Elnakib included 34 randomized or quasi-experimental studies published between 2000 and 2019 [9]. The authors concluded that interventions that increased young girls' access to schooling and those that provided cash or other in-kind assets to their families showed the most consistently positive results for slowing child marriage rates. The interventions, which primarily included large scale (e.g., national) conditional cash transfer programs or efforts to improve educational and economic opportunities for girls, seem to hold great promise. However, they are also very expensive for governments to fund and sustain, particularly those in resource-limited settings with the highest rates of child marriage. Moreover, these interventions do not directly address the gender inequitable beliefs that are a root cause of child marriage. Interventions that provide cash or other material goods that facilitate girls' access to schooling can prevent scenarios like those described above, but they do not necessarily change the underlying gender norms that devalue girls. Recent research shows that gender norms are malleable [12, 13]. For example, an educational intervention among adolescents in India found that classroom discussions about gender equality led to a sustained reduction in support for social norms that devalued women and girls [13].

In order to make meaningful progress toward eliminating child marriage by the year 2030, interventions must be both effective and highly scalable–politically, logistically, and

financially– and on a relatively short timeline. Financial scalability is particularly important at this moment in history because child marriage interventions may compete for public resources being devoted to addressing the ongoing COVID-19 pandemic and its long-term impacts in many domains. Targeted media messaging may be a cost-effective means of changing gender norms. This study will evaluate whether radio programming designed to address highly localized motivations for child marriage can modify attitudes toward the practice in Ibadan, Nigeria.

## Child marriage in Nigeria

Nigeria is home to the largest number of married girls in Africa [14]. National rates of child marriage have remained stubbornly high in recent decades despite domestic and international efforts to address the issue [15, 16]. According to the 2018 Demographic and Health Survey, 43% of Nigerian women between 20 and 24 years of age married before their 18th birthdays [17]. However, the frequency of child marriage varies widely across the country and is much higher in the northern and north eastern regions, where the prevalence is greater than 70% in some states [17]. These geographic differences in child marriage are correlated with state-level laws regarding the practice, ethnicity, and violent conflict.

Twenty-eight of 36 Nigerian states have adopted the federal *Child Rights Act*, which makes marriage before the age of 18 illegal [18, 19]. All states that have thus far refused to adopt the *Act* are located in the north [19], meaning that child marriage is not legally prohibited where it is most commonly practiced. Government representatives in these states have used religious and cultural arguments to justify their opposition to the *Act* [20, 21]. Child marriage rates also vary markedly across ethnic groups in Nigeria. A recent study based on a nationally representative sample found that 55% of Hausa girls between the ages of 15 and 19 were married, a prevalence more than ten times as high as that among the other two major ethnic groups in the country (Yoruba (3%) and Igbo (4%)) and twice as high as among minority ethnic groups in Northern Nigeria [22]. These ethnic differences are strongly correlated with the geographic variation already discussed: the northern states of Nigeria account for the largest concentration of persons of Hausa ethnicity living in the country. In addition, many of these states have been affected by violent conflict for several decades, resulting in substantial migration of Hausa persons from northern to southern regions of the country [23]. Both conflict and migration are likely to exacerbate the drivers of child marriage. Where conflict involves an increased risk of sexual violence, marriage may be seen as a means of protection against sexual assault, thereby increasing the incentive for girls to marry at very young ages. Migration may lead to economic hardship and families may arrange the marriage of young girls to lessen the financial burden of caring for them [24]. In the scenarios described above, it is likely that decisions regarding the marriage of young girls are made by adult members of their families and not by the girls themselves, or at least not exclusively [25]. For these reasons, our intervention will target Hausa adults.

## Methods

### Study setting

The study will be conducted in two cities in southwest Nigeria: Ibadan and Akure. Ibadan will serve as the intervention site and Akure as the control site. Ibadan was selected as the intervention site because one of the principal investigators of this project (Olubukola Omobowale) is a community physician with strong professional networks throughout the city and institutional support from the College of Medicine at the University of Ibadan. Akure was selected as the control site because it is located in a neighbouring state with similar social, demographic, and

geographic characteristics but is outside the broadcast range of radio stations in Ibadan, which will limit spill over of the intervention and, pragmatically, reduce the logistic burden of collecting data in both sites during a short-term project. Both Ibadan and Akure are located in states that legally prohibit marriage before the age of 18. In addition, both cities receive a substantial number of Hausa immigrants from the northern states of the country.

## Intervention

We will develop a series of radio dramas intended to reach members of the Hausa community aged 40 years and older who reside in Ibadan, Nigeria. Radio is an appropriate means of reaching this population. Many Hausa persons living in Ibadan migrated to Southwest Nigeria from northern regions of the country and reside in informal housing settlements in the city. Television is beyond the financial means of many households, and radio is often listened to as a collective social activity among adults. Recent research demonstrates that radio is an important source of public health information in the country [26].

The programs will include fictional storylines that address gender norms that motivate child marriage and will highlight the harms of the practice. The content of the radio dramas will be informed by qualitative research that is currently underway. Our research team is conducting focus group discussions and key informant interviews with Hausa community members living in Ibadan to understand how child marriage is perceived, its potential benefits and harms, and why it is ongoing. The results of this qualitative research will provide the basis for storylines in the radio dramas that counter gender inequality and address the specific motivations for child marriage within the community. The radio dramas will be professionally scripted and produced by the Shield of Innocence Initiative, a non-governmental organization that specializes in developing social messaging for media outlets, with assistance from the Biomedical Communication Centre of the University of Ibadan College of Medicine [27, 28]. A Local Advisory Committee composed of ten Hausa community leaders from Ibadan, including religious authorities, will review the radio dramas prior to broadcasting to ensure that the content is culturally appropriate and compelling.

We have budgeted to air the radio dramas three times per week on three separate Hausa-language stations based in Ibadan over a three-month period in 2023. Incentives will be provided to promote listenership. At the end of each radio drama, listeners who correctly respond to a question about the content will be entered into a drawing for pre-paid mobile phone credit. Two local Hausa celebrities will also be hired to participate in the dramas to increase their appeal to a broad audience. We verified that the broadcast range of these stations does not extend to Akure, the control city, but some of these stations allow listeners to tune into their programming online. This means that some degree of spill over of the intervention is possible. However, our target population includes persons aged 40 and older, whom we believe are unlikely to seek out online radio broadcasts, and we will look for evidence of spill over in Akure during the follow-up survey.

## Research objectives

We aim to learn whether the targeted radio intervention described above improves knowledge about child marriage and influences support for the practice among adult members of the Hausa community residing in Ibadan, Nigeria. Our specific objectives are:

1. To determine whether targeted radio broadcasts can effectively reach Hausa communities in Ibadan

2. To determine whether the targeted radio broadcasts improve knowledge of the harms of child marriage

3. To determine whether the targeted radio broadcasts change support for child marriage among members of the Hausa community in Ibadan

## Research design

We will measure the impact of our intervention using a controlled, pre-post design. A baseline survey will be conducted in both cities prior to the intervention to measure knowledge and perceptions of child marriage, including the ideal age for marriage among boys and girls, the perceived benefits and harms of marrying early, and knowledge of the minimum age for marriage specified by law. The baseline questionnaire was written in English and will be translated into Hausa and Yoruba, the primary languages spoken among Hausa persons living in Ibadan and Akure.

The radio dramas will be broadcast in Ibadan for a period of three months. Afterward, we will follow up with the same individuals who participated in the baseline survey in both cities and ask them to complete a second interview. The follow-up interview will measure coverage of the intervention, i.e., whether participants heard the radio dramas, and will ask many of the same questions as the baseline interview so that we can measure changes in these outcomes over time.

## Pre-determined outcomes and sample size calculations

During the process of applying for funding for this research, the authors were required to specify a threshold for demonstrating "proof-of-concept" to the funding agency, Grand Challenges Canada (GCC) [29]. (Under GCC's funding model, projects that demonstrate proof-of-concept may be eligible for expansion and longer-term funding). The authors and the funding agency agreed in advance on the outcomes and the minimum relative change in those outcomes that would be considered sufficient evidence to demonstrate that the intervention seems promising, i.e., demonstrates proof-of-concept:

1. A 10% increase in the proportion of participants who can identify at least one harmful consequence of child marriage for girls' health, education, and/or economic opportunities

2. A 20% increase in the proportion of participants who know that marriage before the age of 18 is illegal in Oyo State, Nigeria (where Ibadan is located)

3. A 10% reduction in the proportion of participants who believe that the ideal age for girls to marry is younger than 18

4. A 10% reduction in the proportion of participants who would support the marriage of girls in their families before the age of 18.

These pre-determined outcomes and the specified minimum change in those outcomes guided our sample size calculations. We want to be sure that we have sufficient statistical power to detect the effect sizes described above with a high degree of precision and, ideally, to detect even smaller effects. Assuming that 20% of persons surveyed at baseline already have the outcome of interest, we estimate that with a sample of 1600 persons, 50% of whom will be assigned to the treatment group (i.e., living in Ibadan), we can detect a minimum change of five percentage points in each outcome with 80% power at an alpha level of 0.05 [30].

## Sampling procedure

Participants will be recruited through a multi-stage cluster sampling approach. In the first stage, enumeration areas (called settlements) within known Hausa neighbourhoods (wards) of Ibadan and Akure will be randomly selected from existing lists developed by health care centres in those wards. In the second stage, houses within selected enumeration areas will be randomly selected using lists obtained from the Nigerian National Population Commission (NPC). Multiple households often reside within the same residential building in these cities. Every household in a selected building that includes at least one eligible participant will be invited to participate. If households include more than one eligible participant, one individual will be randomly selected. If buildings selected from the NPC lists are abandoned or not in residential use at the time of the survey, new buildings will be randomly selected to replace them. Using this sampling approach, we will randomly select a total of 1600 persons to participate: 800 in Ibadan and 800 in Akure.

## Inclusion criteria

Eligible participants must be 40 years of age or older, be of Hausa ethnicity, reside in one of the selected neighbourhoods in Ibadan or Akure, and have no intention of moving away from the neighbourhood within six months following the baseline survey. Eligible persons must also be willing to share their contact information so that they can be reached for a follow-up interview.

## Variables to be measured

The baseline survey will record demographic characteristics of participants to determine how exchangeable intervention and control samples are prior to the intervention. The baseline and follow-up surveys will also collect information on participants' knowledge and perceptions of child marriage. These questions will be worded identically on both surveys to facilitate measurement of change following the intervention. The baseline survey questionnaire is included in the appendix. We have several outcomes of interest. Those necessary for demonstrating proof-of-concept to the funding agency are:

1. The proportion of persons in Ibadan who report having listened to at least one radio drama during the intervention period. This will be assessed using the question, "Did you hear stories about child marriage broadcast on [radio stations] in the past few months?"

2. The proportion of persons who are aware that marrying before the age of 18 is illegal in Oyo and Ondo state, where Ibadan and Akure are respectively located. This will be measured by correct responses to the question, "According to the law, what is the youngest age at which a person can get married in [Oyo or Ondo] State?"

3. The proportion of persons who believe that the ideal age for girls to marry is younger than 18 years. This will be measured by responses to the question, "In your opinion, what is the ideal age for a girl to get married?"

4. The proportion of persons who would support the marriage of girls in their families before the age of 18, measured based on responses to the question, "Would you support the marriage of a girl in your family before she turned 18?"

5. The proportion of persons who correctly identify at least one harmful consequence of child marriage, as indicated through pre-defined responses to the question, "What are the possible harms to a girl if she marries before she turns 18?"

## Training of data collectors

We will recruit data collectors that speak Hausa and Yoruba to administer baseline and follow-up surveys. Since data collectors will be Hausa-speaking, it is quite likely that they will also be familiar with Hausa culture in southwest Nigeria. Even so, they will be trained to interact politely with elders and to dress appropriately to facilitate the building of trust and rapport with participants. Male and female data collectors will work in pairs to respect gender norms in the Hausa community and maintain the security of our research team. Female participants will be interviewed by female data collectors; male participants will be interviewed by male data collectors.

## Data collection and management

Participants will be asked whether they are most comfortable speaking Hausa or Yoruba and the survey will be conducted in their preferred language. Yoruba is the most commonly spoken language in Southwest Nigeria and some Hausa persons may be more comfortable speaking Yoruba. They will be asked to provide informed consent in their language of choice before beginning the baseline survey.

Reponses to survey questions will be entered directly into Research Electronic Data Capture (REDCap) software loaded on electronic tablets [31]. The College of Medicine at the University of Ibadan is a member of the REDCap consortium and is permitted to use this software. The REDCap mobile application can be used for offline data collection, which makes it suitable for use in this study setting where internet connectivity is unstable. The Field Supervisor will ensure that all data collected offline is sent to the REDCap server on a daily basis. Data will be exported from REDCap to Stata version 17 for analyses [32].

Participant's names, cell phone numbers, and residential addresses will be collected so that we can reach them for a follow-up interview. We will also ask each participant to provide the name and phone number of a family member or friend who can be contacted if the participant cannot be reached for a follow-up interview. In that case, we will ask the friend about the participant's status (e.g., whether they are ill or have moved away) so that we can minimize loss to follow-up during the study. After the follow-up interview is completed, we will assign each participant a code number and remove all personally identifying contact information from the data. All contact information for family or friends will also be deleted at that point.

## Statistical analyses

We will measure the impact of our intervention by comparing the change in outcomes between baseline and follow up in the intervention site (Ibadan) with the change in outcomes between baseline and follow up in the control site (Akure). Outcomes will be modelled using the following basic equation:

$$Y_i = \alpha + \beta C_i + \gamma T_i + \delta(C_i * T_i)$$

This model posits that the outcome for an individual $i$ ($Y_i$) is a function of the city they live in ($C_i$) and the time point $T_i$ (baseline or follow-up). The coefficient $\delta$ represents the treatment effect we are interested in. We will use the coefficients obtained from a logistic regression model to estimate the marginal probability of each outcome among the intervention and control samples and subtract these estimates to calculate risk differences.

Demographic covariates measured at baseline may be added to this model to account for any observed differences between intervention and control samples at baseline.

## Status and timeline of the study

This study has received ethical approval from the University of Ibadan College of Medicine Institutional Review Board and the McGill Faculty of Medicine and Health Sciences Research Ethics Board. This protocol was initially submitted to PLOS ONE in August 2022 and peer reviews were received in February 2023. Recruitment of participants for the baseline survey began in February 2023. The radio dramas are currently being developed and we anticipate they will be aired beginning in April 2023.

Anonymized, individual-level data will be made publicly available through the Open Science Foundation (OSF) no later than two years following conclusion of the follow-up interview.

## Discussion

Innovative interventions targeting child marriage that are cost-effective and rapidly scalable are needed in Nigeria. Legislation against child marriage has thus far failed to curb the practice and it remains common, particularly among Hausa girls. We will evaluate whether targeted, culturally appropriate radio programming can change knowledge of and support for child marriage among Hausa adults living in Ibadan, Nigeria. This intervention is innovative for several reasons. First, the ability of stand-alone media campaigns to inform and change attitudes toward child marriage has never been evaluated using an experimental or quasi-experimental research design, according to three systematic reviews of such interventions [9–11]. Second, the intervention is readily scalable and cost-effective compared to other promising interventions such as cash transfer programs or modifying national or regional education policies (e.g., eliminating tuition fees). Third, the intervention targets adults who are influential in making marriage decisions for children in their families. Prior interventions have sometimes targeted girls themselves, an approach that may have limited impact if girls' agency to make decisions about their marriages is restricted.

## Limitations

Funding for this study has been provided for a one-year period, which severely limits the timeframe for this intervention. The radio dramas will air for a period of three months. Earlier educational interventions that had a positive effect on gender equitable attitudes ran for more than two years [12, 13]. The limited timeframe to evaluate the intervention means that we cannot estimate its effect on actual behaviours like the incidence of child marriage. Instead, we will measure support for those behaviours, which may be a precursor to them. Our intervention may not be able to bring about a measurable change in attitudes over such a short period. However, we have powered our study to be able to detect relatively small effects, which would still be interesting for two reasons. First, if our short-term intervention results in small changes in the outcomes of interest, it is plausible that a longer-term intervention would result in stronger effects. Second, even a small effect from such a short-term and cost-effective intervention may be worth fielding more broadly among populations with high rates of child marriage.

## Acknowledgments

We are grateful to Bidemi Nelson of the Shield of Innocence Initiative for her thoughtful feedback on our ideas and for contributing to the development of the radio dramas.

## Author Contributions

**Conceptualization:** Olubukola Omobowale, Alissa Koski.

**Formal analysis:** Olubukola Omobowale, Alissa Koski.

**Funding acquisition:** Olubukola Omobowale, Alissa Koski.

**Investigation:** Olubukola Omobowale, Alissa Koski.

**Methodology:** Olubukola Omobowale, Alissa Koski.

**Project administration:** Olubukola Omobowale.

**Resources:** Olayinka Egbokhare, Alissa Koski.

**Supervision:** Olubukola Omobowale, Olayinka Omigbodun, Alissa Koski.

**Validation:** Olubukola Omobowale, Alissa Koski.

**Visualization:** Olubukola Omobowale, Olayinka Egbokhare, Alissa Koski.

**Writing – original draft:** Olubukola Omobowale, Alissa Koski.

**Writing – review & editing:** Olubukola Omobowale, Olayinka Omigbodun, Olayinka Egbokhare, Alissa Koski.

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
