## [Decision Letter · Decision Letter 0]

9 Feb 2023

PONE-D-22-23503A Protocol for a Controlled, Pre-Post Intervention Study to Change Attitudes Toward Child Marriage in Southwestern Nigeria Using Targeted Radio ProgrammingPLOS ONE

Dear Dr. Koski,

Thank you for submitting your manuscript to PLOS ONE. After careful consideration, we feel that it has merit but does not fully meet PLOS ONE’s publication criteria as it currently stands. Therefore, we invite you to submit a revised version of the manuscript that addresses the points raised during the review process.

Specifically, please ensure the objectives are clearly stated and all relevant citations are included in the protocol.

We look forward to receiving your revised manuscript.

Kind regards,

Janna Metzler

Academic Editor

PLOS ONE

Journal Requirements:

Reviewers' comments:

Reviewer's Responses to Questions

**Comments to the Author**

1. Does the manuscript provide a valid rationale for the proposed study, with clearly identified and justified research questions?

Reviewer #1: Yes

Reviewer #2: Yes

2. Is the protocol technically sound and planned in a manner that will lead to a meaningful outcome and allow testing the stated hypotheses?

Reviewer #1: Yes

Reviewer #2: Yes

3. Is the methodology feasible and described in sufficient detail to allow the work to be replicable?

Reviewer #1: Yes

Reviewer #2: Yes

4. Have the authors described where all data underlying the findings will be made available when the study is complete?

Reviewer #1: Yes

Reviewer #2: Yes

5. Is the manuscript presented in an intelligible fashion and written in standard English?

Reviewer #1: Yes

Reviewer #2: Yes

6. Review Comments to the Author

You may also provide optional suggestions and comments to authors that they might find helpful in planning their study.

Reviewer #1: It would be helpful to include information on the listenership of the radio stations in Ibadan and then verify access to that radio station in Akure to ensure that there is no contamination or information leak baring in mind the era of internet radio

Reviewer #2: Thank you for inviting me to review this protocol which is timely and designed to address the burning issue of child marriage in Nigeria. These are my comments:

1. The last sentence under financial disclosure is somewhat confusing. Rephrase.

Introduction

2. Provide references for the last two sentences in the second paragraph.

3. Objectives should come at the end of the introduction and not under methods. Also, the design of the radio programmes is also an objective and should be listed as such.

4. Both objectives stated currently are ‘double barreled’. For example, objective 1 seek to determine the effective reach of the radio programmes (an objective on its own) and also determine improvement in knowledge (another objective). I advise that these should be separated.

Methods

5. The study settings should be the first subsection here. The intervention should come after inclusion criteria.

6. The authors repeatedly used the term ‘random selection’ which should be correctly termed ‘simple random sampling’. Simple random sampling is a known sampling method, but random selection is not. This should be corrected appropriately.

Inclusion criteria

7. Individuals of Hausa ethnicity should be added as a criteria (since they are the target) and not just those living in Hausa community, since other ethnic groups, especially the Yoruba hosts also live there.

8. The second and third statements under this section appear contradictory, kindly revisit.

General comments

9. Is there any plan to include Hausa religious and traditional leaders in this study due to their influence in promoting societal norms that affect parents’ choices about the girl child marriage?

10. A mix of British and American English were used in the write up. Choose one of the English versions.

7. PLOS authors have the option to publish the peer review history of their article (what does this mean?). If published, this will include your full peer review and any attached files.

Reviewer #1: No

Reviewer #2: No

---

## [Decision Letter · Decision Letter 1]

4 May 2023

A Protocol for a Controlled, Pre-Post Intervention Study to Change Attitudes Toward Child Marriage in Southwestern Nigeria Using Targeted Radio Programming

PONE-D-22-23503R1

Dear Dr. Koski,

We’re pleased to inform you that your manuscript has been judged scientifically suitable for publication and will be formally accepted for publication once it meets all outstanding technical requirements.

Kind regards,

Janna Metzler

Academic Editor

PLOS ONE

Additional Editor Comments (optional):

Reviewers' comments:

Reviewer's Responses to Questions

**Comments to the Author**

1. Does the manuscript provide a valid rationale for the proposed study, with clearly identified and justified research questions?

Reviewer #1: Yes

Reviewer #2: Yes

2. Is the protocol technically sound and planned in a manner that will lead to a meaningful outcome and allow testing the stated hypotheses?

Reviewer #1: Yes

Reviewer #2: Yes

3. Is the methodology feasible and described in sufficient detail to allow the work to be replicable?

Reviewer #1: Yes

Reviewer #2: Yes

4. Have the authors described where all data underlying the findings will be made available when the study is complete?

Reviewer #1: Yes

Reviewer #2: Yes

5. Is the manuscript presented in an intelligible fashion and written in standard English?

Reviewer #1: Yes

Reviewer #2: Yes

6. Review Comments to the Author

You may also provide optional suggestions and comments to authors that they might find helpful in planning their study.

Reviewer #1: The authors have not addressed all the points raised in the earlier review such as moving the objectives to introductory section as well as the use of random selection

Reviewer #2: This is a timely and interesting study which will hopefully provide an effective intervention to address child bride problems in Nigeria. Most of the earlier concerns have been addressed.

7. PLOS authors have the option to publish the peer review history of their article (what does this mean?). If published, this will include your full peer review and any attached files.

Reviewer #1: No

Reviewer #2: No

---

## [Editor Report · Acceptance letter]

8 May 2023

PONE-D-22-23503R1 

A Protocol for a Controlled, Pre-Post Intervention Study to Change Attitudes Toward Child Marriage in Southwestern Nigeria Using Targeted Radio Programming 

Dear Dr. Koski:

I'm pleased to inform you that your manuscript has been deemed suitable for publication in PLOS ONE. Congratulations! Your manuscript is now with our production department. 

Kind regards, 

on behalf of

Dr Janna Metzler 

Academic Editor

PLOS ONE